# Apigenin: A Therapeutic Agent for Treatment of Skin Inflammatory Diseases and Cancer

**DOI:** 10.3390/ijms24021498

**Published:** 2023-01-12

**Authors:** Ji Hye Yoon, Mi-Yeon Kim, Jae Youl Cho

**Affiliations:** 1Department of Biocosmetics, Sungkyunkwan University, Suwon 16419, Republic of Korea; 2School of Systems Biomedical Science, Soongsil University, Seoul 06978, Republic of Korea; 3Department of Integrative Biotechnology, Sungkyunkwan University, Suwon 16419, Republic of Korea

**Keywords:** apigenin, skin inflammation, flavonoid, skin, atopic dermatitis, psoriasis, pruritus, skin cancer, vitiligo

## Abstract

The skin is the main barrier between the body and the environment, protecting it from external oxidative stress induced by ultraviolet rays. It also prevents the entrance of infectious agents such as viruses, external antigens, allergens, and bacteria into our bodies. An overreaction to these agents causes severe skin diseases, including atopic dermatitis, pruritus, psoriasis, skin cancer, and vitiligo. Members of the flavonoid family include apigenin, quercetin, luteolin, and kaempferol. Of these, apigenin has been used as a dietary supplement due to its various biological activities and has been shown to reduce skin inflammation by downregulating various inflammatory markers and molecular targets. In this review, we deal with current knowledge about inflammatory reactions in the skin and the molecular mechanisms by which apigenin reduces skin inflammation.

## 1. Introduction

The skin may be divided into several layers according to their main functions, including the epidermis, dermis, and hypodermis [1]. The epidermis is the outermost layer of the skin and acts as a barrier to prevent disruption by external stimuli [2,3]. Moreover, it acts as a defense system to retain immune homeostasis against many pathogens, including bacteria, antigens, and viruses [4]. The epidermis consists largely of keratinocytes and some Langerhans cells (LC), melanocytes, and Merkel cells [5]. Among these, the Langerhans cell is a type of dendritic cell (DC) that can present the necessary antigen for the innate immune response. When LCs are activated by ultraviolet (UV) irradiation, E-cadherin is expressed at low levels in keratinocytes. As a result, migration of LCs into the lymph nodes occurs, and regulatory T cells can be further matured [6,7]. Idoyaga et al. [8] revealed that skin DCs can be targeted for immunomodulatory therapies. In the outer layers of the epidermis, the skin microbiome retains skin acidity to protect against external infections [9,10,11]. Organisms in the microbiome also interact with each other to form a community in the skin, and these interactions ensure immune homeostasis in the skin. On the other hand, dysregulation of the inflammatory equilibrium can induce skin immunological diseases [12,13,14], including atopic dermatitis (AD), pruritus, psoriasis, skin cancer, and vitiligo [15,16,17]. As part of the efforts to attenuate inflammatory symptoms, steroidal and non-steroidal drugs are mostly used as classical treatments [18,19,20,21,22]. However, because of the side effects of anti-inflammatory drugs, studies investigating natural compounds to replace chemical drugs have been more actively performed [23].

Apigenin (4′,5,7-trihydroxyflavone, Figure 1) is a natural compound that belongs to the subclass of flavonoids [24,25]. In previous studies, *Tanacetum*, *Achillea*, *Artemisia*, and *Matricaria* genera belonging to the *Artemisia* family of plants have been reported as the main source plants of apigenin [26,27,28,29]. It has been presented that apigenin has the potential to attenuate skin inflammatory conditions, such as AD, pruritus, and psoriasis, and the tumorigenic response of skin cancers (Figure 2). In addition, its anti-apoptotic [30], anti-inflammatory [31], and anti-hyperglycemic effects [32] have been researched. In this review, we focus on describing the functional properties of apigenin and its potential for pharmacological effects. 

## 2. Anti-Inflammatory Effects of Apigenin on UV-Irradiated Skin 

UV light is the main cause of skin inflammation and can be divided into UVA (320–400 nm), UVB (280–320 nm), and UVC (100–280 nm), according to the wavelength. Especially, UVA and UVB penetrate the skin layers, and can induce skin inflammation and aging in keratinocytes and fibroblasts [33,34]. UVB exposure causes an acute inflammation response in the epidermis by promoting the synthesis and secretion of pro-inflammatory cytokines, such as tumor necrosis factor (TNF)-α and interleukin (IL)-6 from epidermal cells [35]. Moreover, UVA penetrates the dermal layers and indirectly causes DNA damage with degradation of the collagen and elastin fibers [34].

In previous studies, apigenin attenuated skin inflammation by downregulating the expression of cyclooxygenase-2 (COX-2) [36]. Another finding suggests its potential as a novel target for reducing skin inflammation. This compound works in the keratinocyte by targeting the non-receptor tyrosine kinase (e.g., Src) and COX-2 [37]. Apigenin prevented cyclobutene pyrimidine dimers, which are generated by UV exposure. Due to this, apigenin restored the lower level of nucleotide excision repair proteins and affected cell apoptosis [38,39]. Apigenin downregulated the level of metalloproteinase-1 by interfering with the Ca^2+^ influx-dependent mitogen-activated protein kinase (MAPK) and activator protein-1 (AP-1) pathways in HaCaT or normal human dermal fibroblast cells under UVA-irradiated conditions. Especially, the levels of c-Jun and c-Fos were decreased by apigenin treatment, which suppressed the phosphorylation of extracellular signal-regulated kinase (ERK), c-Jun N-terminal kinase (JNK), and p38 [40,41]. As a result, apigenin could attenuate UV-mediated inflammation by decreasing the transcription of inflammatory cytokines via the downregulation of the AP-1, MAPK, and apoptotic signaling pathways (Figure 3).

## 3. Effect of Apigenin on Attenuating AD

AD is a chronic inflammatory disease that affects 80% of patients in infancy or childhood. Its severe symptoms include itching, dry skin, eczema, and swallowing. The cytokines and chemokines secreted in AD are summarized in Table 1; however, the exact mechanism of the stimulus is still unclear. There are many hypotheses to explain AD pathogenesis, including: (I) disproportion of skin microbiomes [3,42,43,44]; (II) weakness of skin barrier junctions [45,46]; (III) dysregulation between pro- and anti-inflammatory cytokines [47,48]; and (IV) excessive immunoglobulin E (IgE) secretion. Among these explanations, apigenin reduced IgE and interferon (IFN)-γ levels in serum in an NC/Nga mouse model. Moreover, apigenin attenuated damaging skin lesions induced by picrylchloride. Considering the protein levels, apigenin suppressed the phosphorylation of the signal transducer and activator of transcription 6 (STAT6) in IL-4–stimulated mouse spleen cells [49]. In addition, apigenin showed a low expression of IL-31 in messenger RNA in a human mast cell line (HMC-1). In HMC-1 cells, apigenin downregulated nuclear factor-κB (NF-κB) pathway proteins, including the inhibitor of κB kinase, inhibitor of κB, and the p65/NF-κB, and MAPK pathway factors of c-Jun N-terminal kinase, ERK, and p38 [50]. In summary, it is thought that apigenin can ameliorate the symptoms of AD by decreasing the levels of pro-inflammatory cytokines and inflammatory mediators by downregulating the MAPK, NF-κB, and Jak/STAT signaling pathways (Figure 4).

## 4. Treatment with Apigenin for Alleviating Pruritus

Pruritus or itching negatively affects quality of life [65,66]. Many different factors play a role in this event, but the exact pathogenetic mechanisms are not known [67,68]. Histamine, serotonin, cytokines, peptides, and phospholipid metabolites are included as mediators of pruritus [69]. Among these, cytokines are strong players that manage itching by activating receptors [70]. One example, IL-31, which is derived from the IL-6 family, acts as the therapeutic target of pruritus in the Th2 cell-mediated response. IL-31 has been reported to induce chemokines such as CCL1, CCL17, and CCL22 [71,72]. Additionally, IL-33 has also been considered a pathophysiologically important cytokine that manages innate immune responses [73] and Th2 cell differentiation by promoting the expression of chemokines and pro-inflammatory cytokines, and by activating natural killer cells and dendritic cells [74]. Apigenin weakened the expression of IL-31 in human mast cells and mouse skin through downregulation of MAPK and NF-κB signaling [50]. In an ovalbumin-induced BALB/c mouse model, apigenin regulated the balance of Th1/Th2 cells by downregulating the NF-κB pathway and reducing histamine, IgE, and STAT1 expression. Moreover, apigenin improved the Th1 response by controlling the expression of IFN-γ and T-box protein expressed in T cells [75]. Furthermore, apigenin-treated microglial cells lowered the expression levels of IL-31 and IL-33 without displaying cytotoxicity. This expression was verified by polymerase chain reaction as well as Western blotting via the inhibition of the ERK and JNK pathways [76]. In experiments using astrocytes, apigenin significantly suppressed IL-31 and IL-33 messenger RNA expression. Pre-treatment with apigenin in astrocytes decreased the expression levels of IL-31 and IL-33 at the protein level. In astrocytes, apigenin also inhibited the phosphorylation of MAPK and NF-κB signaling proteins [77]. Taken together, these studies suggest that apigenin can ameliorate pruritus by inhibiting IL-31 and IL-33 secretion through suppression of the NF-κB and MAPK pathways (Figure 5).

## 5. The Mechanism of Apigenin for the Amelioration of Psoriasis

Psoriasis is a chronic, immunological skin disease affecting about 125 million patients in America [78]. These patients and others motivate the study of psoriasis to improve pathophysiological knowledge of the condition. Psoriasis is a sustained inflammatory disease caused by the hyperproliferation of keratinocytes and dysfunctional differentiation. In addition, the infiltration of Th17 cells secreting inflammatory cytokines, such as IL-23, into keratinocytes, dominantly occurs in psoriasis [79]. 

Apigenin showed the greatest effects in a psoriasis model by decreasing cytokine levels. Skin barrier recovery effects were observed in apigenin-treated skin. Apigenin also improved the skin’s condition by increasing the hydration level of the stratum corneum. Meanwhile, apigenin influenced the synthesis of skin structural proteins such as filaggrin, involucrin, and loricrin in mouse models [80]. With co-treatment of apigenin and lipopolysaccharide (LPS) in DCs, this compound significantly inhibited TNF-α messenger RNA expression. Moreover, apigenin suppressed the level of pro-inflammatory cytokines, including IFN-γ, IL-6, IL-1β, IL-23, and IL-10, in both LPS treatment and non-treatment groups. Meanwhile, apigenin affected naïve T cell differentiation by modulating the function of DCs [81]. Overall, apigenin treatment may ameliorate psoriasis symptoms by regulating the transcription of inflammatory cytokines via regulation of the Toll-like receptor 4 pathway (Figure 6).

## 6. The Suppressive Activity of Apigenin on Skin Cancer

The inflammation response promotes cell proliferation to renew damaged cell tissues, so it plays a pivotal role in retaining tissue homeostasis [82]. However, chronic inflammation is also known to induce tumorigenesis. The tumor microenvironment is initiated by the excessive production of inflammatory cytokines. Thus, this phenomenon merits study. Many cytokines and chemokines can be induced in hypoxic conditions in this tumorigenic environment [83]. Skin cancer is a malignant tumor, particularly in Caucasians, with about 1 million cases occurring annually in the United States [84,85,86]. Skin tumors have been named according to their involved cells and clinical behavior. There are three types of skin tumors: basal cell carcinoma; cutaneous malignant melanoma (CM); and non-melanocytic skin cancer (NMSC), which is also known as squamous cell carcinoma [84,87]. Chronic UV exposure most commonly leads to skin cancer among all known risk environments and affects gene mutation, immunosuppression, and oxidative stress [88,89,90]. From various studies, there are several approaches to suppress skin cancer, including through the PI3K/Akt/mTOR, TNF-related apoptosis-inducing ligand, JAK/STAT, and MAPK signaling pathways [91,92,93,94]. 

### 6.1. NMSC

Apigenin downregulates the Akt signaling pathway in UVB-irradiated keratinocytes, blocking the mammalian target of rapamycin (mTOR) activation and suppressing the cell cycle and cell proliferation in mouse skin and keratinocytes. Meanwhile, it promotes autophagy via mTOR inhibition, which inhibits keratinocyte proliferation [95,96]. In primary human epidermal keratinocytes and a skin cutaneous squamous cell carcinoma cell line (COLO-16), treatment with apigenin decreased the conversion of the microtubule-associated protein 1 light chain 3 (LC3) and GFP-LC3 puncta [97]. Apigenin also inhibited skin carcinogenesis by downregulating the COX-2 expression level in UVB-irradiated mouse skins [98]. It is commonly known that 12-*O*-tetradecanoylphorbol-13-acetate (TPA) can induce a tumor by binding and activating the protein kinase C signaling pathways [99,100]. Apigenin treatment suppressed PKC activity dose-dependently and inhibited TPA-mediated carcinogenesis in mouse skin [101,102]. In summary, it is speculated that apigenin treatment can reduce tumorigenic responses by inducing autophagy and via the inactivation of Akt and PKC in keratinocytes (Figure 7).

### 6.2. CMs

Apigenin inhibited the proliferation of melanoma cell lines by downregulating the AKT signaling pathway, which promotes cell apoptosis [103]. Moreover, apigenin treatment suppressed melanoma metastasis to the lungs in C57BL/6 mice and inhibited the phosphorylation of STAT3 in melanoma cells [104]. The presence of apigenin induced anti-melanoma effects by triggering the apoptosis of A375SM cells. Apoptotic proteins, including the caspases, p53, Bcl-2-associated X protein, and poly ADP-ribose polymerase (PARP), were upregulated by apigenin treatment. The compound also downregulated the levels of Akt, STAT3, and MAPK in melanoma cells [105]. Therefore, it is suggested that apigenin could inhibit carcinogenesis by inducing apoptosis in melanoma cells and downregulating the activities of some important survival factors, such as STAT3, Akt, and MAPK proteins (Figure 8).

## 7. The Therapeutic Effects of Apigenin on Vitiligo

Vitiligo has been reported as a pigmentary disorder impacting about 1% of the world’s population [106]. The symptoms of this disease include inconsistent and various sizes of white spots found on the skin and a change of hair color to white [107,108,109]. The main cause of vitiligo is concerned with the autoimmune chronic destruction of melanocytes. The death of pigment cells expands the white lesions on the skin. Therefore, the therapeutic strategy has focused on preventing the apoptosis of melanocytes from oxidative stress and suppressing the proinflammatory response [110]. Of many different compounds, apigenin was reported to upregulate antioxidant enzyme activities, such as superoxide dismutase (SOD), catalase (CAT), and glutathione peroxidase (GSH-Px). Moreover, apigenin was found to promote the gene expression level that is involved in the antioxidant process the at mRNA and protein levels [111]. Meanwhile, it was revealed that apigenin can affect dopamine (DA)-triggered apoptosis in melanocytes by downregulating cleaved PARP and cleaved caspase 3 levels [112]. Apigenin also protected melanocytes from apoptosis by blocking the phosphorylation of Akt, p38, and JNK, which are induced by DA [112] (Figure 9).

## 8. Conclusions

This review described the attenuating effects of apigenin on skin inflammatory conditions and cancer as summarized in Table 2. A natural flavonoid, apigenin showed the greatest activity by attenuating the symptoms of skin inflammatory diseases and tumorigenic responses. Apigenin downregulated inflammatory cytokine expression by suppressing the AP-1, MAPK, and NF-κB pathways in keratinocytes. In addition, apigenin induced autophagy by decreasing mTOR activity and inactivating Akt and PKC activities. Moreover, apigenin protects the cell from oxidative stress-induced cell death. Through treatment, apigenin could prevent skin inflammatory responses to retain the proper regulation of inflammatory cells. Furthermore, apigenin could affect the synthesis of skin barrier factors and Ca^2+^ influx. Based on this review, apigenin could be applied to treat skin inflammatory diseases and cancer.

## 9. Perspective

Apigenin is a bioactive compound used as a therapeutic agent for various diseases, such as diabetes, Alzheimer’s disease, cancer, and amnesia [113,114,115,116]. Treatment with apigenin has led to decreased levels of many inflammatory cytokines. According to study results, apigenin could inhibit the inflammatory response in the skin by downregulating transcription factors, such as AP-1, NF-κB, and STAT. These mechanisms not only back up the excellence of apigenin, but also suggest the possibility of using it as a drug for inflammatory skin diseases. There are now many trials assessing active natural compounds as substitutes for chemically synthetic drugs because of severe side effects associated with the latter [117,118]. Previous studies administered apigenin by applying it to damaged skin or cells to attenuate skin inflammation. Nowadays, apigenin products manufactured with chamomile extracts are being sold in markets to attenuate stress hormones, and bad dreams, as well as in the form of commercially available capsules that provide powerful antioxidant supplements to promote healthy aging and skin health [119]. However, flavonoids can be degraded by high temperatures [120], thus studies aimed at stabilizing apigenin therapeutics should be performed. We suggest that apigenin can not only be used as a therapeutic material, but also as a health supplement for skin diseases, based upon the various studies discussed in this review.

## Figures and Tables

**Figure 1 ijms-24-01498-f001:**
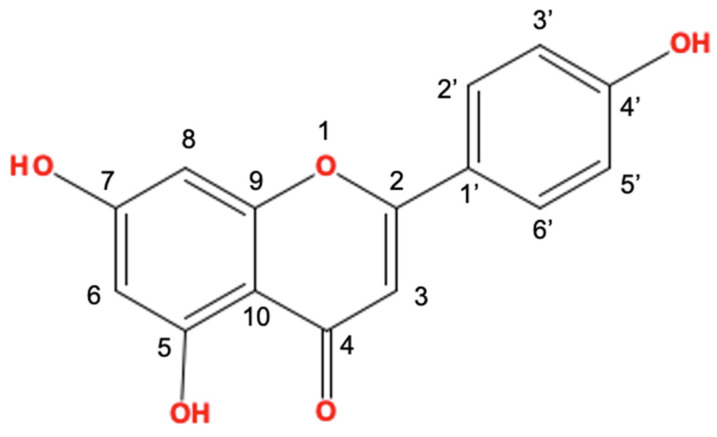
The structure of apigenin (4′,5,7-trihydroxyflavone). This figure was made using JChemPaint software.

**Figure 2 ijms-24-01498-f002:**
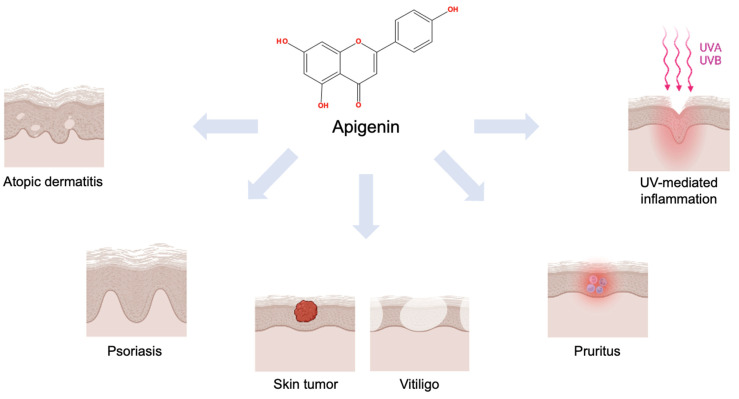
Scheme of the therapeutic efficacies of apigenin on the skin. This figure was created with BioRender.com.

**Figure 3 ijms-24-01498-f003:**
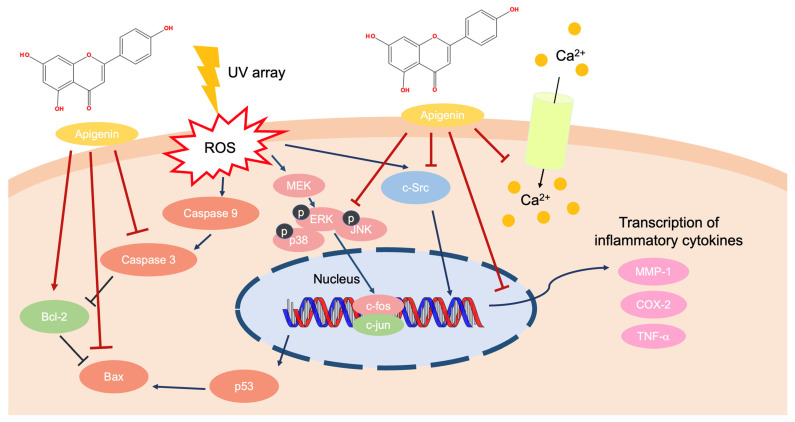
The molecular interactions of apigenin due to UV irradiation.

**Figure 4 ijms-24-01498-f004:**
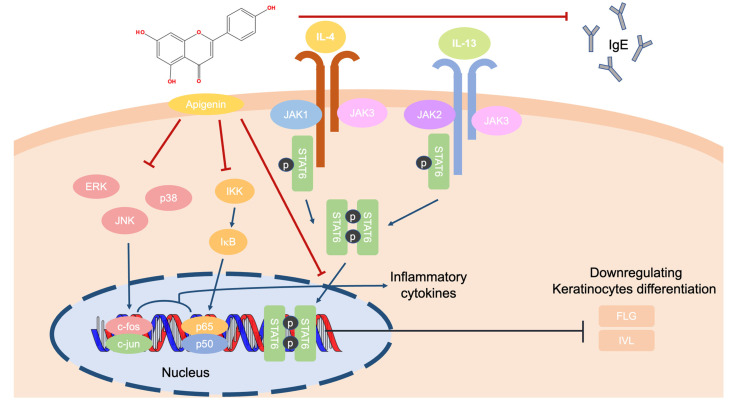
The role of apigenin in protein regulation in AD-irritated skin cells.

**Figure 5 ijms-24-01498-f005:**
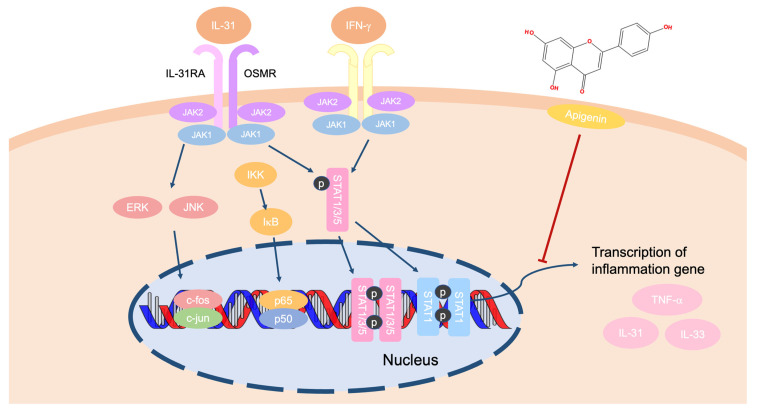
A schema of the efficacy of apigenin against pruritus.

**Figure 6 ijms-24-01498-f006:**
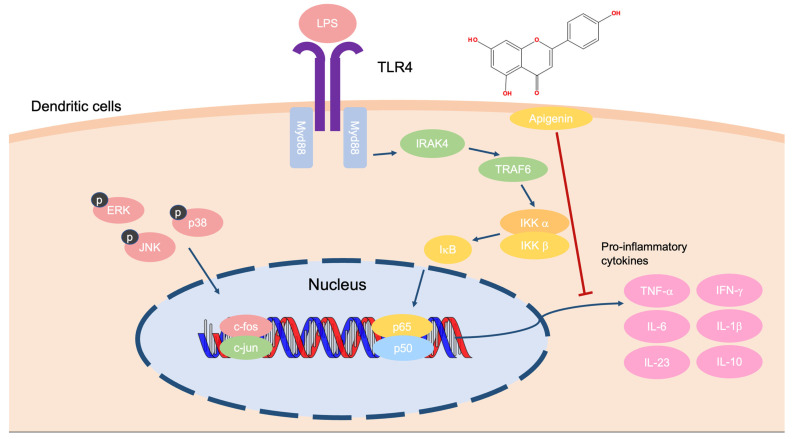
The mechanism of apigenin in LPS-treated DCs.

**Figure 7 ijms-24-01498-f007:**
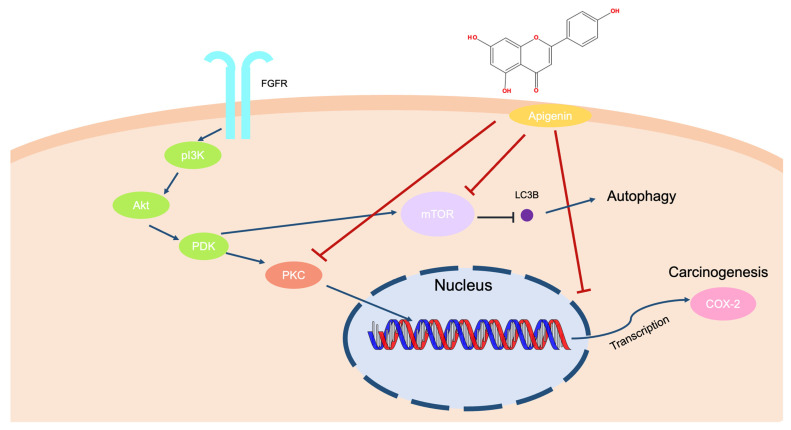
Mechanism of anti-carcinogenesis effects according to apigenin treatment in an NMSC model.

**Figure 8 ijms-24-01498-f008:**
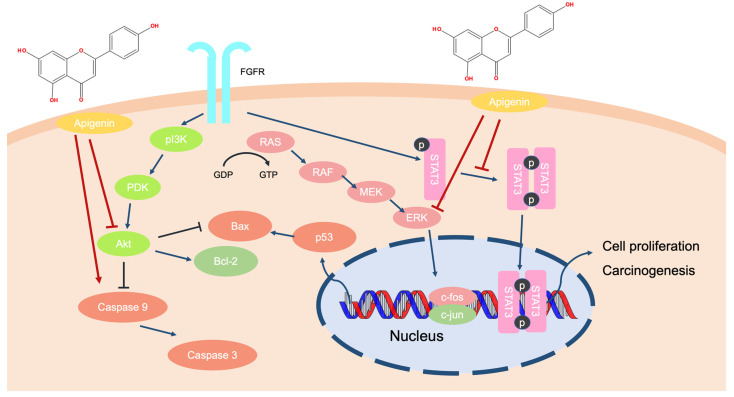
The molecular process of apigenin in melanoma cells.

**Figure 9 ijms-24-01498-f009:**
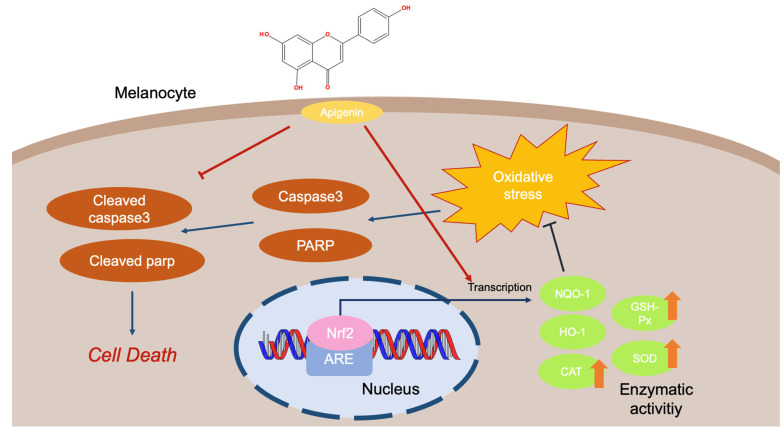
Antioxidant response of apigenin to protect melanocytes against oxidative stress.

**Table 1 ijms-24-01498-t001:** The cytokines and chemokines that stimulate AD.

Cytokines	Classification	Role	Reference
IL-4	Th2 cell-derived cytokines	Interacts with keratinocytes to produce eosinophil chemokine (CCL26) in the acute phase. Regulates IgE secretion from B cells. Directly acts on itch sensory neurons to promote pruritus.	[51,52]
IL-13			
IL-31	Th2 cell-derived cytokines	Causes pruritus by binding the IL-31 receptor A (IL31RA). Downregulates barrier-associated protein expression. Inhibits keratinocyte differentiation.	[53,54]
IL-1α	IL-1 family	Recruits leukocytes to inflammation sites. Decreases the natural moisturizing factors in stratum corneum layers.	[55,56]
IL-1β			
IL-33	IL-1 family	Regulates the activation of mast cells, ILC2, and basophils. Secretes pruritic cytokines from Th2 cells or keratinocytes.	[57,58,59]
IFN-γ	Th1 cytokines	Recruits CXCR3 agonistic chemokines, which induce the T cell into the inflammation site.	[60]
TNF-α			
TSLP	IL-7-like cytokines	Promotes dendritic cells to differentiate into naïve T cells. Binds to the TSLP receptor which is placed in cutaneous sensory neurons to induce pruritus.	[61,62,63,64]

**Table 2 ijms-24-01498-t002:** Mechanism of apigenin against inflammatory skin diseases and cancers.

The Type of Skin Disease	Mechanisms of Apigenin	Test Model	Dose	References
UV-mediated inflammation	Downregulates Src and COX-2 levels.	In vitro	10, 20, 40, 50 μM	[36,37]
	Regulates the level of apoptotic proteins and anti-apoptotic proteins.	In vitro	7, 15 μM	[38,39]
	Inhibits MMP-1 expression by suppressing Ca^2+^ influx.			[40,41]
	Suppresses the MAPK and AP-1 signaling pathways.	In vitro	1, 5, 10, 20 μM	
Atopic dermatitis	Suppresses phosphorylation of STAT6 in IL-4 stimulated mouse spleen cells.	Ex vivo	25 μM	[49]
	Ameliorates damaged skin lesions induced by picrylchloride(piCl).	In vivo	0.05% feed to mice	
	Downregulates the protein levels of the NF-κB, MAPK pathways.	In vitro	10, 20, 30 μM	[50]
Pruritus	Suppresses IL-31 levels by inhibiting the NF-κB and MAPK signaling pathways.	In vitro	10, 30 µM	[50]
	Regulates Th1/Th2 balance by inhibiting the NF-κB pathway, and levels of histamine, IgE, and STAT1 expression.	In vivo	5, 10, 20 mg/kg of mice	[75]
	Enhances the Th1 response by decreasing the expression of IFN-γ, and T-box proteins in T cells.			
	Shows low expression of IL-31, IL-33 in apigenin-treated microglial cells via downregulating ERK and JNK expression.	In vitro	5, 10, 20, 40, 60, 80, 100 µM	[76]
	Inactivates MAPK and NF-κB proteins.	In vitro	30, 60 µM	[77]
Psoriasis	Promotes the synthesis of skin barrier factors.	In vivo	60 µL of 0.1% apigenin in 100% ethanol	[80]
	Downregulates the mRNA expression of inflammatory cytokines in LPS-treated DCs.	In vitro	8, 20 μM	[81]
Skin cancer	Downregulates mTOR and AKT signaling pathways.	In vitro	25 μM	[95,96]
		In vivo	5 μM in 0.2 mL DMSO/acetone (1:9) vehicle mix of mice	
	Induces autophagy by inhibiting mTOR expression and the conversion of LC3.	In vitro	6, 12, 25, 50 μM	[97]
	Decreases carcinogenesis in TPA-mediated mouse skin and PKC activity,.	In vivo	5, 25 μM to mice 10, 50, 100 μM	[101,102]
		In vitro		
	Attenuates melanoma metastases to the lung by decreasing STAT3 levels.	In vivo	150 mg/kg of mice	[104]
	Promotes the expression of apoptotic proteins in A375SM cells. Inactivates the Akt and MAPK pathway proteins.	In vitro	25, 50, 75, 100 µM	[103,105]
		In vivo	25, 50 mg/kg of mice	
Vitiligo	Promotes antioxidant enzyme activity in dose-dependent ways. Increases the expression of antioxidant genes at the mRNA and protein levels.	In vitro	1, 5, 10, 20 µM	[111]
	Protects pigment cells from DA-induced apoptosis by decreasing the level of apoptotic agents.	In vitro	10 µM	[112]
	Inactivates p38, JNK, and Akt levels in the presence of DA.			

## Data Availability

The data are contained within the article.

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
