# Peer review of "Apigenin: A Therapeutic Agent for Treatment of Skin Inflammatory Diseases and Cancer"

_ijms, 2023, doi:10.3390/ijms24021498_

Round 1
Reviewer 1 Report
My main comments:
1/ keywords cannot duplicate with the title
2/ names of chapters and subchapters should be capitalized
3/ on the structural formula, please mark individual rings in the flavonoid skeleton and number the carbon atoms
4/ "in vitro", "in vivo", etc. please write in italics
5/ please adapt the method of citation to IJMS requirements - especially citations where the names of given authors appear in the text
6/ in the names of chemical compounds, the oxygen atom "O" must be written in italics
7/ for the drawings/graphics used, the source should be given
8/ Figure 9. please adjust the font to match the one used throughout the text
9/ please provide DOI numbers in the References section
10/ please provide confirmation of the linguistic correction of the text
Reviewer 2 Report
1. Figure 9 content is very poor. it must be redrawn.
2. application of apigenin in different skin disease must be more elaborated.
3. Dose of apigenin for different skin disease must be reported.
4. A summarized and comprehensive table must be added for the apigenin application in different disese.
5. Is there any marketed product of apigenin reported.
6. Figure resolution must be improved.
Round 2
Reviewer 1 Report
Accept in present form.
Reviewer 2 Report
Accept